# Evidence for Ecosystem-Level Trophic Cascade Effects Involving Gulf Menhaden (*Brevoortia patronus*) Triggered by the *Deepwater Horizon* Blowout

**Jeffrey W. Short** [1,*], **Christine M. Voss** [2], **Maria L. Vozzo** [2,3], **Vincent Guillory** [4], **Harold J. Geiger** [5], **James C. Haney** [6] **and Charles H. Peterson** [2]

1 JWS Consulting LLC, 19315 Glacier Highway, Juneau, AK 99801, USA
2 Institute of Marine Sciences, University of North Carolina at Chapel Hill, 3431 Arendell Street, Morehead City, NC 28557, USA; c.m.voss.unc@gmail.com (C.M.V.); Maria.Vozzo@gmail.com (M.L.V.); cpeters@email.unc.edu (C.H.P.)
3 Sydney Institute of Marine Science, Mosman, NSW 2088, Australia
4 Independent Researcher, 296 Levillage Drive, Larose, LA 70373, USA; vinceguillorysr@yahoo.com
5 St. Hubert Research Group, 222 Seward, Suite 205, Juneau, AK 99801, USA; geiger@ak.net
6 Terra Mar Applied Sciences LLC, 123 W. Nye Lane, Suite 129, Carson City, NV 89706, USA; jchrishaney@terramarappliedsciences.com
* Correspondence: jwsosc@gmail.com; Tel.: +1-907-209-3321

**Abstract:** Unprecedented recruitment of Gulf menhaden (*Brevoortia patronus*) followed the 2010 *Deepwater Horizon* blowout (DWH). The foregone consumption of Gulf menhaden, after their many predator species were killed by oiling, increased competition among menhaden for food, resulting in poor physiological conditions and low lipid content during 2011 and 2012. Menhaden sampled for length and weight measurements, beginning in 2011, exhibited the poorest condition around Barataria Bay, west of the Mississippi River, where recruitment of the 2010 year class was highest. Trophodynamic comparisons indicate that ~20% of net primary production flowed through Gulf menhaden prior to the DWH, increasing to ~38% in 2011 and ~27% in 2012, confirming the dominant role of Gulf menhaden in their food web. Hyperabundant Gulf menhaden likely suppressed populations of their zooplankton prey, suggesting a trophic cascade triggered by increased menhaden recruitment. Additionally, low-lipid menhaden likely became "junk food" for predators, further propagating adverse effects. We posit that food web analyses based on inappropriate spatial scales for dominant species, or solely on biomass, provide insufficient indication of the ecosystem consequences of oiling injury. Including such cascading and associated indirect effects in damage assessment models will enhance the ability to anticipate and estimate ecosystem damage from, and provide recovery guidance for, major oil spills.

**Keywords:** oil spill; forage fish; trophic cascade; marine; food web modeling

## 1. Introduction

Despite widely held assumptions, demonstrating ecosystem-level damage from large marine oil spills remains challenging. Difficulties in clearly establishing ecosystem-level effects include the high natural variability of affected populations, inadequate or absent baseline data, and poorly understood ecological linkages. Nonetheless, when it occurs, damage from ecosystem-level effects may be irreversible and irreparable, and the harm inflicted has the potential to exceed damage at the organismal or population levels of ecological organization [1].

Mass mortalities of vulnerable species that are major components of marine ecosystems provide one of the more likely sources of ecosystem-level effects following large environmental perturbations such as oil spills. Species such as seabirds and marine mammals that routinely occupy or traverse the air–water interface are particularly vulnerable

to direct contact with oil, as are species that inhabit the intertidal zone [2]. These species may also be vulnerable to additional mortality from contact with dispersants, or the from effects of shoreline clean-up efforts [2]. Long-term studies of shorelines oiled by the 1989 *Exxon Valdez* oil spill in Prince William Sound, Alaska, detected persistent damage to oiled intertidal communities, initiated by acute mass mortalities of *Fucus* sp. seaweed, herbivorous limpets and periwinkles, and predatory *Nucella* sp. snails. This resulted in chthamaloid barnacle density increases to far above reference levels 2.5-years after the spill, and was a phenomenon that occurred across hundreds of kilometers of shorelines [3]. This is an example of an indirect ecosystem-level effect following the initial mortalities of *Fucus* sp., limpets, periwinkles and snails from direct oiling.

Another instance of an indirect ecosystem-level effect likely resulted from the 2010 *Deepwater Horizon* (DWH) blowout in the Gulf of Mexico (GoM). The DWH blowout occurred on 20 April 2010, releasing 4.9 million barrels of oil, and was the largest oil spill in United States history to date [4]. Mass mortalities from the direct oiling of seabirds, shorebirds and marine mammals, along with freshwater diversions of Mississippi River waters into coastal marshes, substantially reduced predation on age-0 juvenile Gulf menhaden (*Brevoortia patronus*), contributing to an unprecedented increase in recruitment to age-1 subadults the following year [5,6]. Population biomass during the middle of the fishing season ranged from 0.88 to 1.5 million t from 1977 through 2009, but increased to 2.4 million t in 2011 followed by 1.6 million t in 2012 [6]. Recruitment of the 2010 year class was concentrated near the Mississippi River delta, where coastal oiling was most intense and persistent [6]. Gulf menhaden migrate little along the coast interannually [7], so the increased biomass in 2011 and 2012 was likely concentrated near the Mississippi River delta, particularly west of the river in and near Barataria Bay, resulting in disproportionately large effects on the food web of this region.

The magnitude of the Gulf menhaden biomass, largely confined to the nearshore waters of the northern GoM, suggests that this species may be a major trophic intermediary between the nearshore phyto- and zooplankton they consume, and the wide variety of piscine, mammalian, and avian predator species that consume them (e.g., [8–10]). Species that occupy this role in marine food webs have been described as "wasp-waist" forage species (sensu Bakun [11]). Large perturbations of wasp-waist forage fish populations can have substantial effects on the populations of species they consume, or that are consumed by them [11].

Our objectives here are first to present evidence showing that the physiological condition of Gulf menhaden (i.e., weight-at-length) declined abruptly and in concert with increased recruitment after the DWH blowout. We next compare the annual productivity of Gulf menhaden with the total productivity available to support all species feeding at the same trophic levels of juvenile, sub-adult and adult Gulf menhaden. This comparison supports the hypothesis that Gulf menhaden is a wasp-waist species that experienced nutritional stress as a result of the high recruitment, and subsequently reduced food availability, after the DWH blowout. Finally, we discuss the ecological implications of our findings, and the consequences for modeling food web perturbations triggered by mass mortalities of seabirds and other vulnerable apex predators. This provides a foundation for anticipating such effects after future spills, and for guiding critical data collection in anticipation of, or immediately after, an oil spill incident.

## 2. Methods

### 2.1. Study Area, Gulf Menhaden Life History and Fishery

Our results are based mainly on samples of Gulf menhaden collected by or from commercial seine vessels that operated out of the Daybrook Fisheries, Inc. (DFI) port and processing facility at Empire, Louisiana near the Mississippi River delta (Figure 1). Similar facilities located at Cameron and Intracoastal City, Louisiana, and at Moss Point, Mississippi, were operated by a different company which, along with the DFI facility, account for the entire commercial reduction fishery for Gulf menhaden since 2000. A third

company operated a separate processing facility at Empire, Louisiana from 1964 to 1991. As part of their reduction fishery port sampling program, the US National Marine Fisheries Service (NMFS) collected samples from each processing facility throughout its commercial operation beginning in 1964 to support assessments of the Gulf menhaden stock. The Gulf menhaden fishery is the second largest in the US by volume, with catches near 500,000 t since 2000, of which 26% to 43% was caught by DFI. Most of DFI's catch is from waters in the vicinity of the Mississippi River delta, from Breton Sound in the east to Vermillion Bay in the west (Figure 1).

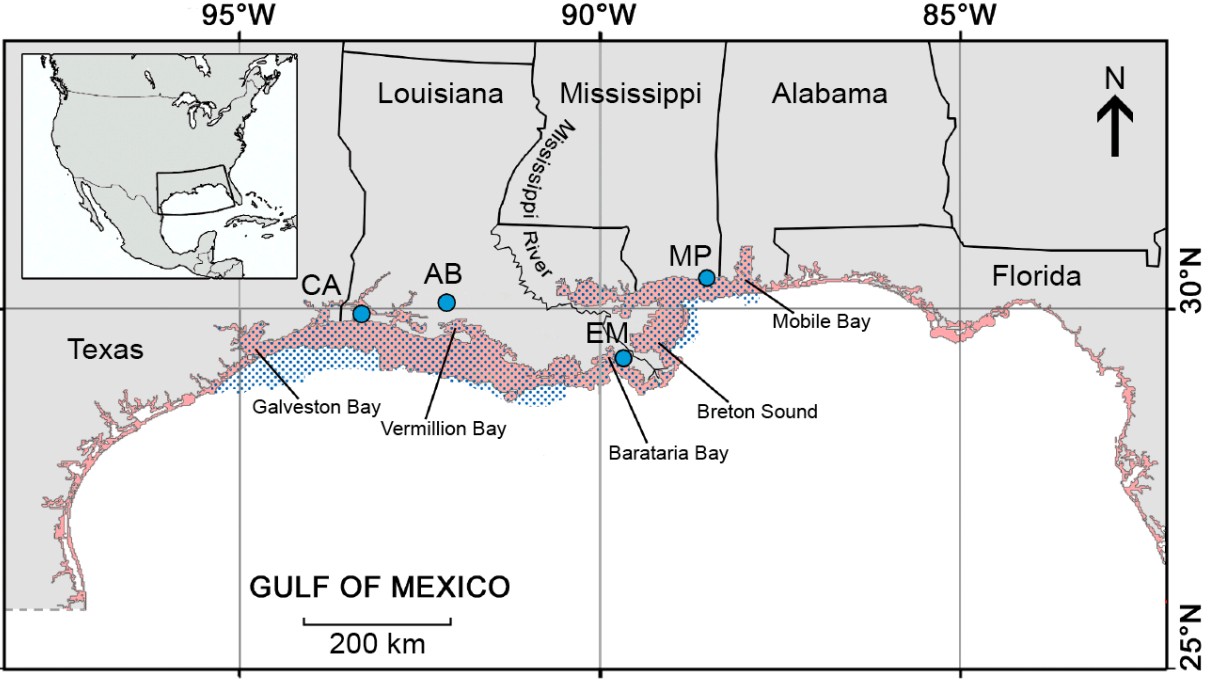

**Figure 1.** Coastal waters inhabited by Gulf menhaden in the United States. Red shading indicates the spatial distribution of commercially exploitable aggregations of Gulf menhaden in the northern Gulf of Mexico, as indicated by commercial purse seine sets during 1986–2011 (from Figure 5.11 in [8]), and represents an area of ~40,000 km$^2$. Blue shading indicates the area assumed inhabited by 75% of the population (see Results section). Blue dots show locations of processing plants at Cameron (CA), Abbeville (AB), Empire (EM) and Moss Point (MP).

The geographical range of Gulf menhaden within the coastal GoM is from the Yucatan Peninsula to the west coast of Florida. Most of the population inhabits estuaries and neritic waters of ≤20 m depth [12], and most of their estuarine nursery habitat is located on the coasts of Louisiana and Mississippi [8]. Gulf menhaden spawn offshore from September through April, with January 1 of the spawning season identifying the age-class year. After wind-driven advection to the upper marshes of coastal estuaries, larvae metamorphose to age-0 juveniles, forming schools in more open waters through the spring, summer and early fall [13]. Juveniles feed primarily on detritus and phytoplankton, switching to mostly phyto- and zooplankton as age-1+ subadults and adults [14]. Gulf menhaden mature near the end of their second year of life as age-1 fish and participate in the offshore reproductive migration. Age-1 fish are partially vulnerable to the fishery when it commences near the end of April, becoming fully vulnerable the following year at age 2. Gulf menhaden may live to age 7, but rarely survive past age 4 [8]. Although older adult Gulf menhaden tend to move towards the Mississippi River delta [7], the river presents a substantial barrier to mixing between sub-populations to its east and west [15].

### 2.2. US National Marine Fisheries Service Port Sampling

Beginning in 1964, the NMFS has measured the length, wet weight and age of Gulf menhaden sampled from the catch of randomly selected commercial fishing vessels at processing plants during the April–October fishing season [8]. We retrieved these data from NMFS for the DFI processing plant located at Empire for the years 1964 through 2012, except for 1970 and 1971, which were not included in the response to our data request to NMFS, and 1977, when the plant processed no fish. We were not able to retrieve comparable data for processing plants at other locations because of data confidentiality restrictions imposed by the US Magnuson-Stevens Fishery Conservation and Management Act (MSFCMA).

### 2.3. Daybrook Fisheries Sampling

Alerted by anecdotal reports from DFI purse seine vessel captains soon after the commercial fishery opened in April 2011 that Gulf menhaden seemed unusually abundant but in poor condition, we implemented an opportunistic sampling plan beginning in May 2011. The objective of this plan was to evaluate quantitatively the spatial and temporal variation of Gulf menhaden individual body condition in the harvest, reasoning that any large change in the characteristics of the harvest would necessarily reflect a change in the characteristics of the population that supports the harvest. The Louisiana coast was partitioned into 10 contiguous sectors. Captains employed by DFI were asked to collect a sample of Gulf menhaden from each of the first three purse seine hauls they set within every sector they fished in each week; compliance was variable. Each sample of Gulf menhaden consisted of 30 fish that were collected ad libitum from each purse seine haul as it was brought aboard the fishing vessel. The location, date and time was recorded for each sample. About 60 of these samples each containing ~30 fish were analyzed from near the beginning, middle and end of the fishing season, resulting in ~5400 individual fish length and weight measurements each year from 2011 through 2014.

Sampled fish were kept refrigerated for up to 5 days aboard fishing vessels, and stored at −20 °C within 48 h of return to port. After storage times of less than a year, fish lengths and weights were measured immediately after thawing. A separate experiment to evaluate the effects of freezing and thawing on wet weight measurements for frozen storage periods of 1 day to 8 months showed that average wet weight declined by 1.47% without apparent trend over this period.

### 2.4. Gulf Menhaden Condition Index and Oil Content

Gulf menhaden lengths and wet weights were measured immediately after thawing to determine body condition using Le Cren's condition index $K_n = w/\hat{w}$ [16], where $w$ is the measured fish weight and $\hat{w}$ is the weight predicted from the length, as $\hat{w} = aL^b$, $L$ = fork length (mm). The constants $a$ and $b$ were set to $8.13 \times 10^{-6}$ and 3.18, respectively, and are derived from the regression of the natural logarithm of wet weight on fork length of 388,831 measurements of fork length and wet weight of Gulf menhaden collected from all ports by NMFS from 1977 through 2011 (results presented in Table 3.9 in [8]). Values of $K_n < 1$ indicate underweight fish. Ages were determined by analysis of annular rings on scales for a randomly selected 16.7% or 20% subsample of each 30-fish sample using NMFS methodology [17,18]. Collection of these data allowed us to evaluate the spatial variation in Gulf menhaden condition in DFI catches for 2011 through 2014, independent of the MSFCMA confidentiality restrictions.

Gulf menhaden oil content was computed as the ratio of annual fish oil produced from the annual DFI commercial fishery catch wet weight.

### 2.5. Biomass, Productivity and Trophic Transfer Computations

We compute Gulf menhaden biomass, $B_y$, for year $y$ as the product of abundance ($N_{a,y}$) and weight ($w_{a,I}$) at the start of the calendar year, summed across age classes $a$:

$$B_y = \sum_{a=1}^{3} N_{a,y} w_{a,I} \tag{1}$$

We use the term "annual production" to denote total production of Gulf menhaden per year, measured in units of g wet weight per year or g carbon (C) per year, and "productivity" to denote production measured in units of g C per year per square meter of sea surface area (i.e., g C/y/m$^2$). We approximated annual production ($P_y$) of Gulf menhaden during year $y$ using an estimated average annual gain in wet weight of each age cohort $a$, summed across ages. For each age cohort, the estimated average annual weight gain was computed as the sum of (1) the product of the model-based estimate of the number of fish that die (i.e., $N_{a,y} - N_{a+1,y+1}$) and their average weight gain at death, $\overline{w}_a - w_{a,I}$, where $w_{a,I}$ and $\overline{w}_a$ are the estimated average weights of an individual age-$a$ fish at the beginning and the middle of the year, respectively; and (2) the product of the number of fish that survive to age $a + 1$ ($N_{a+1,y+1}$) and the average increase in weight from the beginning to the end of the year, $w_{a,F} - w_{a,I}$, where $w_{a,F}$ is the estimated average weight at the end of the year. The product of the number of fish that recruit to the juvenile life stage ($N_{0,y}$) and the average weight at recruitment ($w_{0,I}$) was added to the computed productivity for age-0 juveniles. Our estimate of the total annual production ($P_y$) is therefore:

$$P_y = N_{0,y} w_{0,i} + \sum_{a=0}^{2} \left[ \left( N_{a,y} - N_{a+1,\,y+1} \right) \left( \overline{w}_a - w_{a,I} \right) + \left( N_{a+1,\,y+1} \right) \left( w_{a,F} - w_{a,I} \right) \right] \tag{2}$$

The estimated annual abundances-at-age are from Table 7.3 in [8], except for age-0 juveniles in 2010, which we assume was 170 billion fish instead of 270 billion fish for reasons given in [6]. Fixed weights-at-age at the beginning ($w_{a,I}$), middle ($\overline{w}_a$) and end ($w_{a,F}$) of a year are from Table 3.10 in [8]. These values for weights- and abundances-at-age are listed in Tables 1 and 2, respectively, and are used with Equations (1) and (2) to compute biomass in terms of g wet weight and annual production in terms of g wet weight/y in Table 2. Annual production in terms of g C/y, after converting wet weight to carbon mass using conversion factors of 0.5661 g dry weight/g wet weight and 0.334 g C/g dry weight [19], are presented in Table 3. We did not use data from the most recent Gulf menhaden stock assessment for reasons presented in the supplemental material.

**Table 1.** Weight-at-age of Gulf menhaden at the beginning ($w_{a,I}$) and end ($w_{a,F}$) of a year, and the average over a year ($\overline{w}_a$), from Table 3.10 in [8].

| | Weight-at-Age (g) | | | |
|---|---|---|---|---|
| **Age:** | 0 | 1 | 2 | 3 |
| $w_{a,I}$ : | 0.5 | 45.3 | 95.9 | 145.9 |
| $w_{a,F}$ : | 45.3 | 95.9 | 149.9 | 187.9 |
| $\overline{w}_a$ : | 10.6 | 69.6 | 121.6 | 168.1 |

**Table 2.** Abundance-at-age data used with Equations (1) and (2) (see Methods section) to compute biomass (B), the production (P) of age-0 and age-1+ Gulf menhaden, and the annual ratio of production to biomass (P/B) from 2000 through 2012. Boldface data indicate the 2010 year class. Italics indicate data estimated from Figure 7.11 in [8] for year 2012, and from these estimated 2012 age-0 and age-1 abundances multiplied by $e^{-Ma}$ for year 2013, where $M_a$ is the natural mortality rate of 1.62 for age-0 and 1.3 for age-1 fish from Table 3.11 in [8]. Note that $10^9$ g = 1000 t.

| Year | Abundance-at-Age ($10^9$ Fish) 0 | 1 | 2 | 3 | Biomass, B ($\times 10^9$ g Wet wt) | Annual Production, P ($\times 10^9$ g Wet wt/y) Age 0 | Age 1+ | Sum | P/B |
|---|---|---|---|---|---|---|---|---|---|
| 2000 | 99 | 22 | 5.3 | 0.02 | 1500 | 1700 | 810 | 2500 | 1.7 |
| 2001 | 100 | 20 | 5.3 | 0.09 | 1400 | 1800 | 750 | 2600 | 1.9 |
| 2002 | 80 | 20 | 4.7 | 0.16 | 1400 | 1400 | 730 | 2100 | 1.5 |
| 2003 | 99 | 16 | 4.5 | 0.15 | 1200 | 1700 | 600 | 2300 | 1.9 |
| 2004 | 89 | 20 | 3.6 | 0.15 | 1300 | 1600 | 700 | 2300 | 1.8 |
| 2005 | 83 | 18 | 4.4 | 0.03 | 1200 | 1400 | 670 | 2100 | 1.8 |
| 2006 | 130 | 16 | 4.2 | 0.14 | 1200 | 2300 | 600 | 2900 | 2.4 |
| 2007 | 120 | 27 | 3.9 | 0.08 | 1600 | 2100 | 930 | 3000 | 1.9 |
| 2008 | 47 | 24 | 6.3 | 0.10 | 1700 | 820 | 920 | 1700 | 1.0 |
| 2009 | 100 | 9.3 | 6.1 | 0.64 | 1100 | 1800 | 470 | 2300 | 2.1 |
| 2010 | **170** | 20 | 2.4 | 0.45 | 1200 | **3700** | 680 | 4400 | 1.7 |
| 2011 | 110 | **54** | 4.5 | 0.03 | 2900 | 1900 | **1700** | 3600 | 1.9 |
| 2012 | 120 | 22 | **12** | 0 | 2100 | 2100 | 1000 | 3100 | 1.5 |
| 2013 | | 24 | 5.9 | 0 | | | | | |
| 2000–2009 Average: | | | | | 1400 | 1700 | 720 | 2400 | 1.8 |

**Table 3.** Annual production of age-0 and age-1+ Gulf menhaden from Table 2 converted to units of grams carbon per year (g C/y) and grams carbon per year per square meter of sea surface (left-hand columns), and expressed as a proportion of total productivity available at the trophic levels of age-0 and age-1 fish, and their sum (right-hand columns). Conversion to g C/y/m$^2$ is based on the assumption that 75% of the Gulf menhaden productivity occurs within the blue shaded area of Figure 1, which represents 44,500 km$^2$ (see Methods).

| Year | Annual Production $\times 10^9$ g C/y Age 0 | Age 1+ | Productivity g C/y/m$^2$ Age 0 | Age 1+ | % Mean Productivity Available at Trophic Level of: Age 0 | Age 1+ | Sum |
|---|---|---|---|---|---|---|---|
| 2000 | 320 | 150 | 5.4 | 2.5 | 8.6 | 13 | 22 |
| 2001 | 340 | 140 | 5.7 | 2.4 | 9.2 | 12 | 21 |
| 2002 | 270 | 140 | 4.5 | 2.4 | 7.3 | 12 | 19 |
| 2003 | 320 | 110 | 5.4 | 1.9 | 8.6 | 9.4 | 18 |
| 2004 | 300 | 130 | 5.1 | 2.2 | 8.1 | 11 | 19 |
| 2005 | 270 | 130 | 4.5 | 2.2 | 7.3 | 11 | 18 |
| 2006 | 440 | 110 | 7.4 | 1.9 | 12 | 9.4 | 21 |
| 2007 | 400 | 180 | 6.7 | 3.0 | 11 | 15 | 26 |
| 2008 | 160 | 170 | 2.7 | 2.9 | 4.3 | 15 | 19 |
| 2009 | 340 | 90 | 5.7 | 1.5 | 9.2 | 7.7 | 17 |
| 2010 | **700** | 130 | **12** | 2.2 | **19** | 11 | 30 |
| 2011 | 360 | **320** | 6.1 | **5.4** | 9.7 | **27** | 37 |
| 2012 | 400 | 190 | 6.7 | 3.2 | 11 | 16 | 27 |
| 2000–2009 Average: | 320 | 140 | 5.3 | 2.3 | 8.6 | 12 | 20 |

We used two different approaches to estimate productivity. First, we assumed that 75% of the Gulf menhaden population resides within the 20 m isobath along the northern GoM from the southwestern end of Galveston Bay, Texas in the west to the eastern end of Mobile Bay, Alabama in the east (blue shaded area in Figure 1). This assumption is justified by the facts that most of the Gulf menhaden population resides within the

20 m isobath [12], most of the estuarine nursery habitat is along the coasts of Louisiana and Mississippi [8], Gulf menhaden tend to migrate towards the Mississippi River delta as they age [7], and nearly all of the commercial fishing grounds are located within the 20 m isobath from Galveston Bay, Texas to Mobile Bay, Alabama (see Figure 5.11 in [8]). We estimated the surface area of the blue shaded area in Figure 1 by comparing the number of pixels inside it (83,578 pixels) with the number of pixels within a square of 200 km sides on the same map (79,076 pixels), which resulted in an estimated area of 44,530 km$^2$. We then assumed that 75% of the Gulf menhaden population that we assumed resides within the 20 m isobath from Galveston Bay, Texas to Mobile Bay, Alabama accounts for 75% of Gulf menhaden productivity. This assumption is justified because Gulf menhaden found outside the blue-shaded area in Figure 1 do not contribute disproportionately to productivity. We estimated productivity as the ratio of 75% of annual production in terms of g C/y computed from Equation (2), and the estimated size of the blue shaded area (44,500 km$^2$) in Figure 1. Second, we estimated the Gulf menhaden productivity in terms of g C/y/m$^2$ that contributes to the fishery catch biomass, and the locations where these catches occurred. Regulatory reporting requirements provide accurate estimates and locations of Gulf menhaden catches beginning in 1964 (see Table 4.4 and Figure 5.11 in [8]. Catches during the decade prior to 2010 averaged 490 thousand t/y, equivalent to $9.3 \times 10^9$ g C/y/m$^2$, from a sea surface area of about 40,000 km$^2$ (red shaded area, comprising 75,170 pixels, in Figure 1).

We estimated total productivity at trophic level $i$ as the product of net primary production $P_1$, expressed as g C/y/m$^2$, and the trophic transfer efficiency $\varepsilon$ raised to the $i - 1$ power (see [20]):

$$P_i = P_1 \varepsilon^{i - 1} \tag{3}$$

We assumed an annualized primary productivity of 600 g C/y/m$^2$ along the GoM coast near the Mississippi River delta, based on daily measurements presented in Figure 3 in [21], and a single trophic transfer efficiency of 25% (i.e., $\varepsilon = 0.25$) for each trophic level from primary production to the trophic levels of juvenile and adult Gulf menhaden [9,19,22]. Trophic levels of age-0 and age-1+ Gulf menhaden based on stable nitrogen isotope analyses are estimated as $i = 2.63$ and $i = 3.46$, respectively [14], and are used in Equation (3) to estimate total productivity at trophic level $i$.

### 2.6. Data Quality and Parameter Uncertainty

The NMFS sampling design allocated equal effort across landing ports, fishing vessels and time, providing an approximate random sample of the Gulf menhaden catch. The average numbers of Gulf menhaden measured annually by NMFS for length, weight and age at the DFI port (presented in Figure 2) are 1299 (range 133 to 2631) for age-1 fish, 916 (range 151 to 2353) for age-2 fish, and 123 (range 3 to 400) for age-3 fish. Standard errors of the mean across all ages and years are less than 0.1 g. The accuracy of age determinations has been estimated as ~90% for age-1 fish and ~80% for older fish [23].

The average number of Gulf menhaden measured for each sampling month and location with respect to the Mississippi River for the determination of the Le Cren's condition index ($K_n$; presented in Figure 3) is 823 individuals (range 598 to 1077). Standard errors of the mean were less than 0.003.

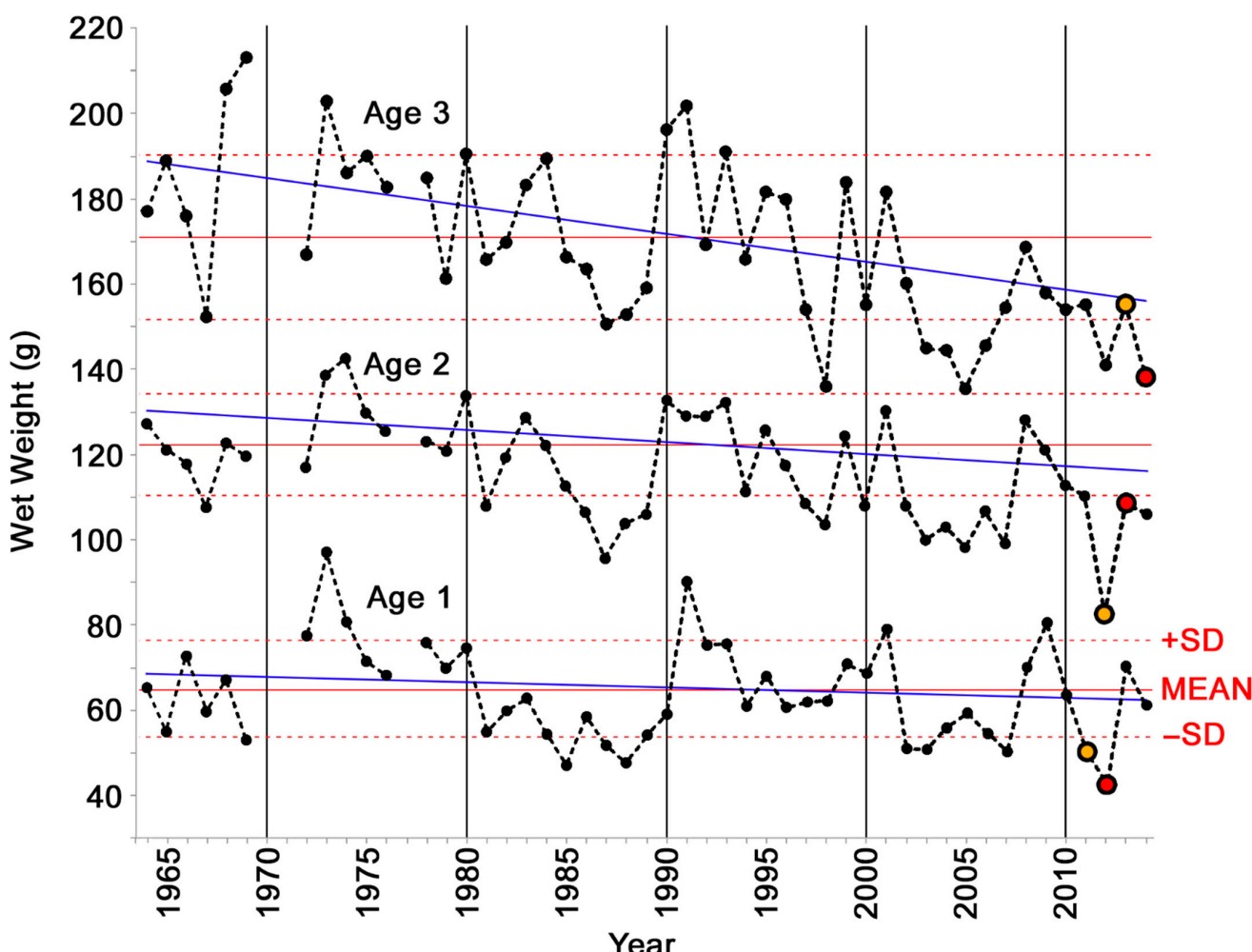

**Figure 2.** Weight of age-1, age-2 and age-3 Gulf menhaden sampled at the Daybrook Fisheries, Inc. port at Empire, LA for the National Marine Fisheries Service port sampling program, 1964 to 2014. Regressions (blue lines) based on 1964 = year zero are: age 1, wet wt. = 68.1–0.134 × year ($p$ = 0.31, $df$ = 42); age 2, wet wt. = 130–0.328 × year ($p$ = 0.015, $df$ = 42); and age 3, wet wt. = 189–0.769 × year ($p$ < 0.001, $df$ = 42). The difference between slopes at ages 2 and 3 are significant ($p$ < 0.001). Orange- and red-filled dots, respectively, indicate the 2010 and 2011 year classes. 95% confidence intervals lie within the black or colored dots.

The uncertainty associated with the Mississippi River flow in May (presented in Figure 4) is likely less than 10% of reported flow [24]. The uncertainty associated with DFI oil yields (presented in Figure 4) is not available but is likely negligible. These oil yields are the ratio of two fixed quantities: the fish oil produced by DFI over the course of a year, and the mass of Gulf menhaden caught and processed to produce that oil. Because the fish oil is sold as an article of commerce, the measurement error of the volume of oil produced is likely close to nil.

The coefficient of variation associated with documentation of catch weights unloaded from individual vessels is estimated as 4% [23]. Summed across hundreds of vessel-unloading events over the course of the fishing season, the uncertainty associated with the total catch weight is likely less by more than an order of magnitude.

The uncertainties associated with the abundances-at-age presented in Table 2 have not been published. The authors of the Beaufort Assessment Model (BAM) used to estimate these abundances assert that simulation testing has shown that the BAM can recover estimated parameters accurately [25], and the database used for the Gulf menhaden assessment is among the best in the U.S. [23]. We therefore take the abundances estimated by the BAM at face value, and assume that the associated uncertainties are small relative to the magnitude of the estimates.

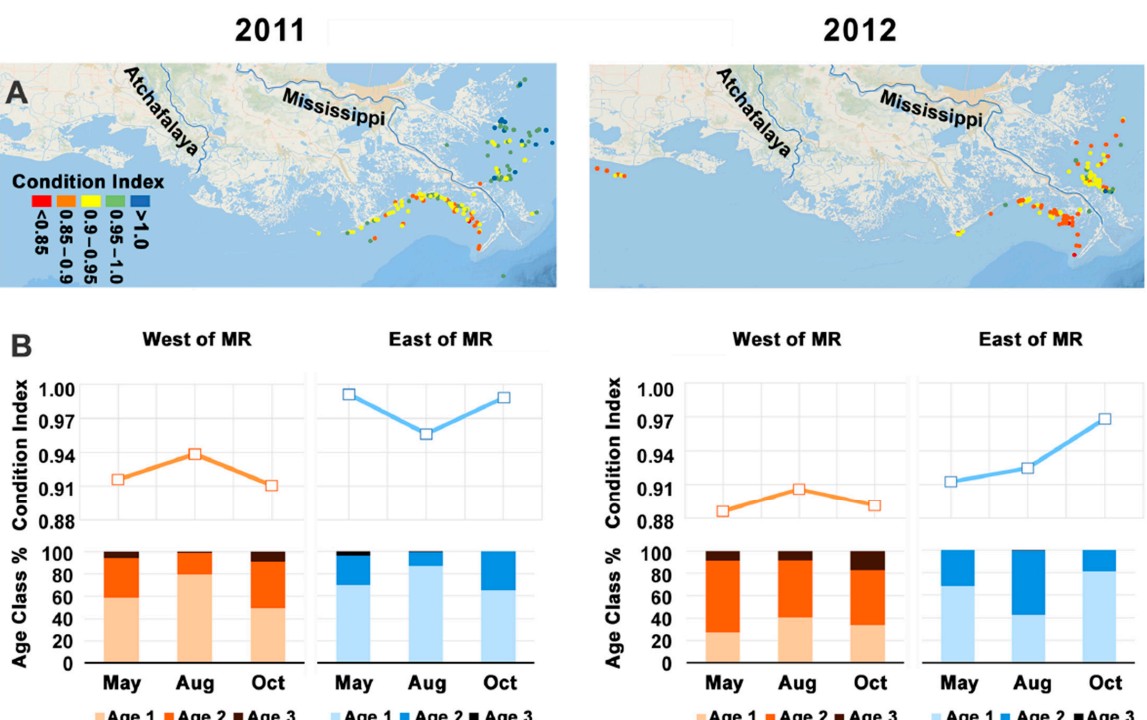

**Figure 3.** (**A**) Map of Gulf menhaden condition index $K_n$ in 2011 and 2012, and (**B**) age-specific mean condition index of samples collected by Daybrook Fisheries, Inc. during May, August and October. Condition index symbols in panel B obscure 95% confidence intervals ($598 \leq n \leq 1077$). MR = Mississippi River.

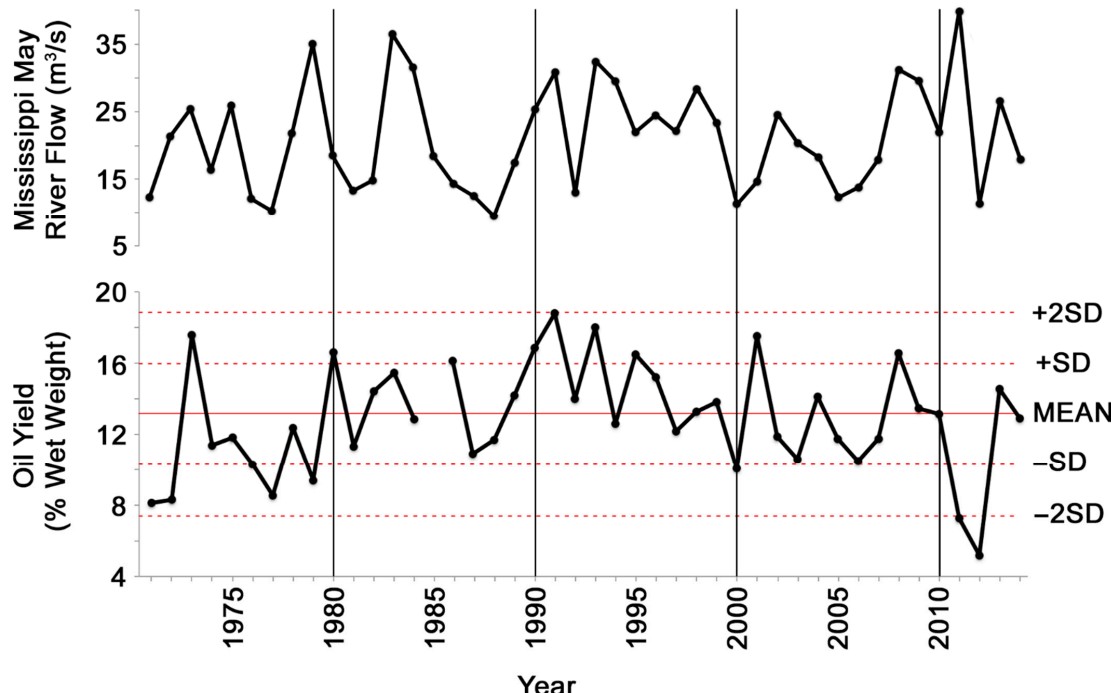

**Figure 4.** Comparison of Mississippi River discharge during May (upper panel) and the annual mean yield of fish oil, as % of wet fish weight for Gulf menhaden processed at Daybrook Fisheries, Inc. reduction plant at Empire, LA, 1971 to 2014.

Our assumed trophic transfer efficiency $\varepsilon$ of 0.25 is likely near the upper limit of feasible values. As noted by Gulland [20], trophic transfer efficiencies are difficult to estimate but rarely exceed 0.25 for marine poikilotherms, and are always less than food

conversion efficiencies because not all production at one trophic level is consumed by the next trophic level. In the northern GoM, losses of ungrazed pelagic primary production to the seafloor sustain the development of an annual hypoxic zone west of the Mississippi River [26], thereby reducing the amount of production available to higher trophic levels, including Gulf menhaden. An increase in our assumed trophic transfer efficiency from 0.25 to 0.30, a value we consider very unlikely, would increase the estimated primary productivity available to age-0 Gulf menhaden from $600 \times (0.25)^{1.63} = 63$ g C/y/m$^2$ to $600 \times (0.3)^{1.63} = 84$ g C/y/m$^2$, an increase of 33%. A similar computation for age-1+ Gulf menhaden leads to an increase of 55%, from 20 g C/y/m$^2$ to 31 g C/y/m$^2$. Conversely, a reduction of the trophic transfer efficiency to 0.20 would decrease the estimated primary productivity available to age-0 and age-1+ Gulf menhaden to 44 g C/y/m$^2$ and 11 g C/y/m$^2$, respectively.

Taking the uncertainties noted above into account, we report computed quantities to two significant figures, but retain the number of significant figures given for published measured parameters used as inputs to Equations (1)–(3).

### 3. Results

The high population biomass of Gulf menhaden in 2011 and 2012 (Table 2) was associated with the exceptionally poor physiological condition near the Mississippi River. Based on the NMFS sampling of commercial landings at the Empire port, the weight-at-age for age-1 fish in 2011 (the 2010 year class) was the third lowest on record since 1964 (Figure 2). At age 2, the weight-at-age of the 2010 year class in 2012 was the lowest on record, at more than three standard deviations below the long-term mean for age-2 fish, indicating the low and declining condition of this year class as the cohort aged. Age-1 fish from the 2011 year class exhibited the worst condition of age-1 fish on record in 2012, nearly two standard deviations below the long-term mean (Figure 2). These two separate year classes (2010 and 2011) of Gulf menhaden, which shared planktonic food resources in the same coastal habitats, were both characterized by exceptionally poor physiological condition in 2012. In 2013 and 2014, the weights-at-age returned to values typical of years prior to 2011 (Figure 2).

The condition of Gulf menhaden collected by DFI seine vessels in 2011 was substantially lower west the Mississippi River delta and in Barataria Bay, the region where coastal oiling from the DWH blowout was most intense and persistent, compared with east of the river (Figure 3 and Figure S1). The condition index ($K_n$) of fish collected west of the Mississippi River during late May–early June 2011 was 0.92, and ranged from less than 0.94 to nearly 0.88 through the end of 2012. East of the Mississippi River, the Gulf menhaden condition (0.95 to 0.98) was only slightly below normal during 2011, but by spring 2012 their condition declined to 0.92 and was nearly as low as the population west of the river, recovering by fall.

The poor physiological condition of Gulf menhaden during 2011 and 2012 around the Mississippi River delta was also associated with unprecedented reductions in commercial fish oil yields. The two lowest mean annual oil yields from fish landed at Empire since the beginning of the record in 1971 occurred in 2011 and 2012, and were more than two standard deviations below the long-term mean prior to 2010. Historically, a pattern of higher oil yields in years with greater flow from the Mississippi River in spring has been evident; however, this trend was reversed in 2011 and 2012 (Figure 4). These low oil yields imply correspondingly low lipid contents of the Gulf menhaden population, because the DFI catches represent such a large proportion of the age-1+ Gulf menhaden population, especially near the Mississippi River. After 2012, the fish oil yields returned to values typical of years prior to 2011 (Figure 4), consistent with the results for weight-at-age from the NMFS port sampling program (Figure 2) and condition indexes based on DFI's sampling program during 2013 and 2014.

During the decade prior to 2010, the annual production of Gulf menhaden was nearly twice its biomass, with age 0 usually accounting for about two-thirds of the annual

production of the stock in a given year (Table 2). On a wet weight basis, biomass in January ranged from 1.1 to 1.7 million t from 2000 through 2009, when the annual production of age-0 fish ranged from 0.82 to 2.3 million t/y and that of age-1+ fish ranged from 0.47 to 0.93 million t/y. Total annual production ranged from 1.7 to 3.0 million t/y, and ratios of production to biomass averaged 1.8, ranging from 1.0 to 2.4. In 2010, total production increased by 43% above the highest value of the prior ten years to 4.4 million t/y, 3.7 million t/y of which was from the age-0 year class, and it increased from an average of 66% of the total production prior to 2010 to 84% of that in 2010. In 2011, total production was 3.6 million t/y, with age-1+ fish accounting for nearly half, at 1.7 million t/y (Table 2).

In 2010, the productivity (g C/y/m$^2$) of age-0 and age-1+ fish accounted for 19% and 11% of available productivity, respectively, their sum being 30%. In 2011 and 2012, the sum was 38% and 27%, respectively. The productivity of age-0 fish increased from an average of 5.3 g C/y/m$^2$ during the decade prior to 2010, to 12 g C/y/m$^2$ in 2010, and this cohort accounted for 93% of the age-1+ productivity of 5.4 g C/y/m$^2$ in 2011 (Table 3). Based on our assumptions for inputs to Equation (3) (see Methods), the total productivity available at trophic levels occupied by age-0 and age-1+ Gulf menhaden was 63 g C/y/m$^2$ and 20 g C/y/m$^2$, respectively. Age-0 and age-1+ Gulf menhaden accounted for averages of 8.6% and 12% of these totals during the decade prior to 2010, implying that their sum, ~20%, of net primary productivity flowed through this species (Table 3).

In comparison, the average removal of 490 thousand t of Gulf menhaden by the commercial fishery from 40,000 km$^2$ of sea surface (see Methods) during the decade prior to 2010 is equivalent to an average biomass of 12 g wet weight/m$^2$, or 2.3 g C/m$^2$. A ratio of productivity to biomass of 1.8 (Table 2) implies a productivity of 1.8 times the biomass of 2.3 g C/m$^2$ per year in these waters, or 4.1 g C/y/m$^2$. About 42% of this biomass was acquired at age 0, while fish were feeding at trophic level 2.63, because the average weight of Gulf menhaden at the end of their first year of life is 45.3 g (Table 1), and the average weight of fish in the commercial fishery catch from 2000 to 2009 was 107.6 g (i.e., 43.5 g/107.6 g = 0.42). The remaining 58% of the biomass of the age-1+ fish in the commercial fishery catch was acquired while fish were feeding at trophic level 3.46. Consequently, annual production of about 0.42 × 4.1 g C/y/m$^2$ = 1.7 g C/y/m$^2$ occurred at the age-0 trophic level of 2.63, and 0.58 × 4.1 g C/y/m$^2$ = 2.4 g C/y/m$^2$ at the age-1+ trophic level of 3.46. The productivity of 1.7 g C/y/m$^2$ at the trophic level of age-0 fish is about 2.7% of the total available productivity of 63 g C/y/m$^2$ at trophic level 2.63, and the productivity of 2.4 g C/y/m$^2$ at the trophic level of age-1+ fish is about 12% of the total available productivity of 20 g C/y/m$^2$ at trophic level 3.46. The sum of these productivities, 15%, is an estimate of the total net primary production flowing through Gulf menhaden based on average fishery catch biomass.

## 4. Discussion

### 4.1. Decline of Physiological Condition of Gulf Menhaden after the Deepwater Horizon Blowout

Three independent lines of evidence imply that the Gulf menhaden population near the Mississippi River delta suffered exceptional nutritional stress during 2011 and 2012. These include: (1) the exceptionally low weights-at-age of the 2010 year class at age 1 in 2011 and at age 2 in 2012, along with the 2011 year class at age 1 in 2012, based on the NMFS port sampling program (Figure 2); (2) the low condition indexes of Gulf menhaden sampled under the opportunistic DFI sampling program (Figure 3); and (3) the low oil yields of DFI catches in 2011 and 2012 (Figure 4). Correspondingly low oil yields of Gulf menhaden catches in 2011 and 2012 by the other fishing company operating in the northern GoM [27,28] indicate that these low yields characterize the entire catch of both years.

The potential factors contributing to the poor physiological condition of Gulf menhaden during 2011 and 2012 include the poor feeding conditions caused by some combination of inadequate food availability and high fish abundance, and toxic effects caused by exposure to crude oil from the DWH blowout. A recent modeling study suggests that oil toxicity may reduce zooplankton abundance [29], but any such reductions would likely



be limited to the year 2010, when the oil contamination of surface waters was widespread in the region. Food availability may have contributed to the poor physiological condition of Gulf menhaden in 2012, but probably not in 2011. Gulf menhaden condition is correlated with Mississippi River discharge [30,31]. Based on the weak but significant ($r = 0.55$, $p = 0.0234$, $df = 16$) correlation between annual oil yield and Mississippi River discharge during May [30], the record-high discharge in 2011 (Figure 4) was expected to generate conditions of abundant nutrients and strong surface water stratification, stimulating planktonic production, thereby promoting above-average oil yields. Instead, the high abundance of Gulf menhaden in 2011 was associated with poor physiological conditions and low fish oil yields (Figures 2–4). However, the low Mississippi River discharge during May 2012 (Figure 4) suggests poor feeding conditions for Gulf menhaden, which likely contributed to the poor physiological conditions of both the 2010 year class at age 2 and the 2011 year class at age 1 (Figures 2 and 3), as well as the low oil yields of 2012 (Figure 4). The more typical May discharges during 2013 and 2014 (Figure 4) imply more normal feeding conditions during those years, consistent with the recovery of physiological condition of the 2010 year class at ages 3 and 4 in 2013 and 2014, and the 2011 year class at age 3 in 2014 (Figure 2). Additionally, Gulf menhaden produced more normal oil yields during 2013 and 2014 (Figure 4).

The toxicity effects from the exposure of Gulf menhaden to DWH crude oil were likely negligible for three main reasons: (1) at the time of the blowout, only a small proportion of the 2010 year class was exposed to oil at age 0; (2) their exposure duration was relatively brief [6]; and (3) little oil from the DWH blowout contaminated the surface waters inhabited by Gulf menhaden after 2010. Similarly, Carroll et al. [32] found relatively minor adverse effects on an Arctic cod fishery in simulated oil spills in their spawning grounds. Milliman et al. [33] suggested the DWH blowout as a possible source of black particulates on the gills, in the stomachs and in the heart tissue of Gulf menhaden they sampled from oiled Barataria Bay and presumably un-oiled Vermillion Bay from 2010 through 2012, but the supporting evidence they present is inconclusive. The total polycyclic aromatic hydrocarbon (PAH) concentrations in whole-body homogenates [34], and the detection frequencies of black particulates and of most histopathological lesions, were not significantly different between the oiled and un-oiled locations from 2010 through 2012. Milliman et al. [33] suggest that the general absence of statistically significant differences between the oiled and reference locations may have resulted from the intermixing of fish, but extensive tagging conducted during the early 1970s [7] indicates this is unlikely. Instead, local or regional combustion sources seem a more plausible source of the black particulates detected by Milliman et al. [33], and if so, the statistically significant interannual variation in total fish PAH concentrations they report may reflect variation in winds and other factors affecting particulate deposition patterns.

The steadily declining weights-at-age of Gulf menhaden prior to 2010 and dating back to 1964 ([35]; Figure 2) suggest that Gulf menhaden had been near the carrying capacity of their habitat prior to the DWH blowout. These declining weights-at-age were not likely the result of selective removals by the Gulf menhaden fishery, because most of the population reproduces before it is vulnerable to the fishery, so selection effects associated with the fishery have scant effects on fitness. As indicated by the slopes of the regression lines in Figure 2, weights-at-age decreased with increasing age, and the increment in slopes from age 2 to age 3 is nearly twice the increment from age 1 to age 2, suggesting that the declining weights-at-age may be the result of declining growth rates. Increasing water temperatures along with decreasing food availability may both contribute to declining growth rates. Higher water temperatures decrease oxygen availability while increasing the metabolic rate of aquatic ectotherms [36], requiring higher food consumption rates to maintain growth rates. The trend of increasing sea surface temperatures by nearly 1 °C since the early 1980s in the northern GoM [37] may therefore have reduced the Gulf menhaden habitat's capacity to support their productivity, thereby increasing their susceptibility to nutritional stress following a large and abrupt increase in recruitment, such as occurred in 2010.

*4.2. Comparison of Gulf Menhaden Productivity with Total Available Productivity*

The increased food consumption required to support the exceptionally productive 2010 Gulf menhaden year class had different effects on the food web in 2010 compared with 2011 and 2012. Age-0 juveniles emerge from coastal marshes about 5 months after hatching to form filter-feeding schools in the more open waters of coastal embayments, consuming mainly phytoplankton [13,14]. Although the increased consumption of mainly phytoplankton by age-0 fish accounted for approximately 19% of available production in 2010 (Table 3), it had little effect on the weight-at-age of older-aged fish during 2010 (Figure 2), or on the 2010 oil yield (Figure 4). This is because the body weight gain of the age-0 fish increased exponentially, starting in mid-spring, so most of the weight gain occurred during the fall. For example, a typical recruitment of 100 billion fish to the juvenile life stage at a body weight of 0.5 g has a biomass of 50,000 t, increasing to 900,000 t at the end of age 0 as numbers decline from ~100 billion to ~20 billion, but body weight increases to 45 g (Table 1). Hence the effects of the increased production of age-0 fish in 2010 on the food web mainly occurred near or after the end of the fishing season, when most of the NMFS port samples had already been collected and most of the catch delivered. In early 2011, the high abundance of age-1 fish from the 2010 year class had a proportionally greater effect on their food supply as they switched to consuming mainly zooplankton [14], nearly a full trophic level higher than that of age-0 fish.

The reasonably close agreement of the two approaches we used to estimate the proportion of net primary productivity that flows through Gulf menhaden offers strong support for the hypothesis that this species plays a dominant role in the food web of their core habitat. Given the relatively high accuracy and precision of the estimates of average catch weight and of the sea surface area from which Gulf menhaden were caught, along with the relatively high values we assumed for primary productivity (600 g C/y/m$^2$) and trophic transfer efficiency (25%), we believe 20% or more of this productivity flowed through Gulf menhaden during the decade prior to 2010 in the waters inhabited by most of the population. Our approximate estimate of 15% of the total net primary productivity appropriated by Gulf menhaden going into the fishery from 2000 to 2009 is not much lower than our estimate of 20% of the total net primary production appropriated by Gulf menhaden based on the assumption that 75% of Gulf menhaden productivity occurs within the 20 m isobath from Galveston Bay, Texas to Mobile Bay, Alabama. Our 15% estimate based on the commercial fishery catch is a minimal estimate, because it does not include production associated with age-0 fish, or with age-1+ fish that were not caught by the fishery, in the waters where fishing occurred during the years 2000 to 2009. Although we have little basis for estimating the contributions of these fish to total Gulf menhaden production in commercially fished waters, the preponderance of the age-0 fish's contribution to total population production (Table 3), along with recognition that ~90% of age-1 fish escape the fishery, suggests that if the production associated with these two groups were included, they might easily increase the estimate of 15% to well above 20%. By comparison, the human appropriation of net primary production globally has been estimated at about 25% [38], suggesting that Gulf menhaden's domination of the food web of their core habitat is comparable to that of human domination of the global ecosystem. This comparison clearly supports identifying Gulf menhaden as a wasp-waist forage fish within their core habitat, although humans are not because they are an apex predator, so their population is not subject to control by predators (excepting micropredators).

Our productivity estimates imply that Gulf menhaden were vulnerable to stress by inadequate food availability in 2011 and 2012. The demand that increased Gulf menhaden production placed on the food supply was likely augmented by production associated with the recruitment increases of other forage fishes in 2010 [39,40] that feed at similar trophic levels. Spatial variation in Gulf menhaden recruitment in 2010 [6] implies that this increased demand was most acute in Barataria Bay and vicinity west of the Mississippi River, such that food demand from Gulf menhaden may have accounted for more than half the available supply there. The intra- and interspecific competition for food that

resulted readily accounts for the evidence of the poor physiological condition of Gulf menhaden we present here. Together, these results strongly suggest that the Gulf menhaden habitat, at least near the Mississippi River, could not supply adequate food to support the increased production associated with the increased recruitment of the 2010 year class of Gulf menhaden at historically normal weights-at-age or lipid content.

*4.3. Ecological Implications*

Gulf menhaden are a typical intermediate-sized forage fish that are critical to sustaining higher trophic-level piscivorous seabirds [10,41–44], marine mammals, and larger predatory fishes [45]. The fisheries literature consistently maintains that over a wide range of forage fish abundances, higher-order predation keeps population size in check; however, when forage fish escape predatory control through some environmental intervention, the population density of forage fish increases dramatically (e.g., [11]). Release from predator control typically leads to exceptionally high abundances, along with poor physiological conditions [11,41,46] resulting from intensified competition for planktonic food resources. Surface-dwelling schools of menhaden are highly visible, often tightly schooling, and thus readily available to seabirds, marine mammals, and the fishery, once located by them [18]. When released from predatory control, density-dependent competition for planktonic foods leads to negative relationships between population size and fish weight and condition (e.g., [47]).

The poor physiological condition of Gulf menhaden in 2011 and 2012, where recruitment was highest in 2010, conforms with the responses expected from the productivity of a single large year class as it progresses as a cohort through the ages of greatest abundance. Vital rates of the Gulf menhaden population indicate that the productivity of a year class will be greatest during the first three years of life (Table 2), becoming evident in fishery catches at age 1. The recovery of the physiological condition of catches in 2013 and 2014 is consistent with the duration of the expected effects from the high recruitment of a single year class in 2010, providing additional supporting evidence for unusually strong recruitment that year.

The high productivity of the Gulf menhaden 2010 year class may have suppressed the recruitment of other species through increased predation on their planktonic life stages. Even at normal levels of abundance, menhaden may significantly affect plankton populations in the waters they inhabit through feeding [48]. At elevated abundance, commensurately higher mortality rates are inflicted on their planktonic prey [14,49], suggesting the possibility of trophically cascading effects (Figure 5). The greater recruitment of a single cohort would tend to reduce phytoplankton abundance through direct consumption of phytoplankton at age 0. However, the increased consumption of zooplankton at ages 1 and 2 in subsequent years would tend to increase phytoplankton abundance, by reducing the consumption of phytoplankton by zooplankton. This suggests that episodic reductions in zooplankton by strong year classes of Gulf menhaden as they pass through age 1 and 2 may contribute to the strength of the annual hypoxic zone that develops west of the Mississippi River delta [26], by suppressing the abundance of zooplankton that would otherwise have consumed a portion of the excessive phytoplankton productivity that generates the biological oxygen demand leading to hypoxia.

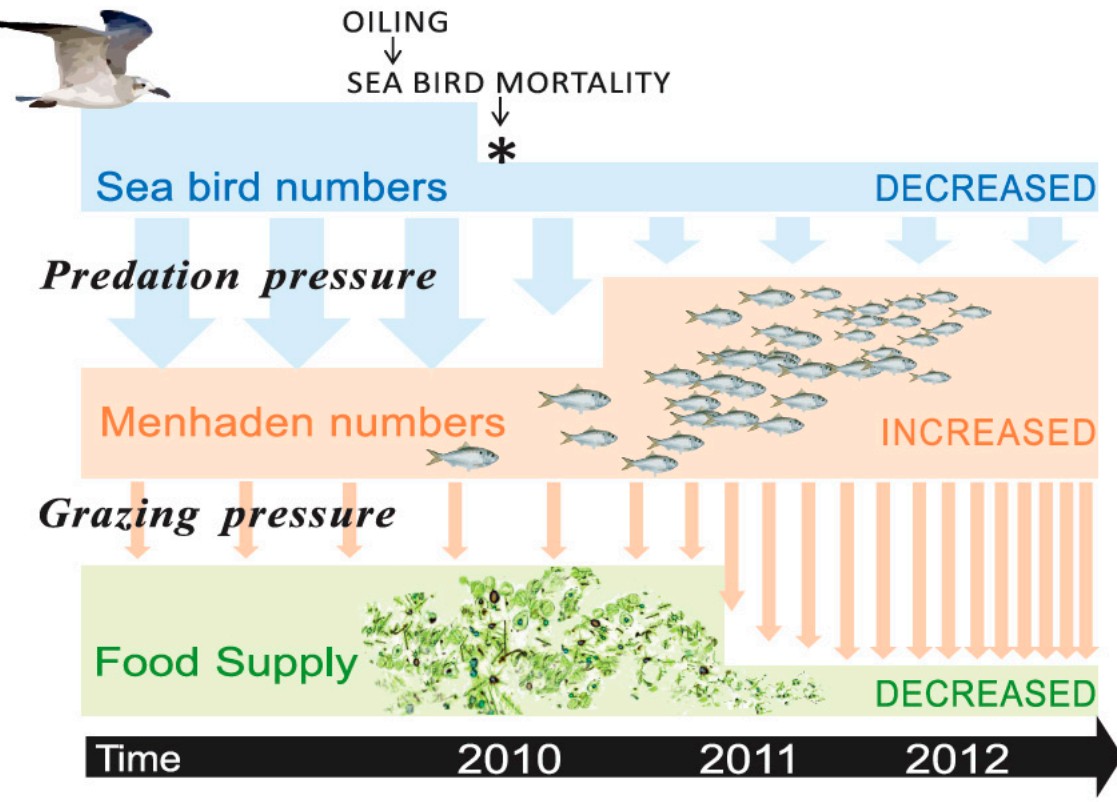

**Figure 5.** Depiction of proposed trophic cascade triggered in part by seabirds killed by oiling from the 2010 *Deepwater Horizon* blowout, increasing recruitment of juvenile Gulf menhaden through reduced predation by seabirds on them, followed by increased consumption of planktonic prey by Gulf menhaden.

If sustained over multiple successive cohorts, the net effect of higher abundance across Gulf menhaden age classes would tend to increase phytoplankton abundance, because of the higher proportion of net primary production that flows through age-1+ fish compared with age-0 fish (Table 3). This difference in the trophic levels of age-0 and age-1+ fish suggests that increased harvests of Gulf menhaden might help reduce phytoplankton abundance. Gulf menhaden recruitment has been almost completely unrelated to spawning stock biomass over at least the last 30 years [8,50], so while moderate increases in catches of age-1+ fish would likely have negligible effects on age-0 abundance and hence their consumption of phytoplankton, the reduced abundance of age-1+ fish would increase zooplankton abundance, which would tend to reduce phytoplankton abundance. Similarly, the release of predation control of sprat in the Baltic Sea has been linked with increased hypoxia there through a similar trophic cascade [51–53]. This suggests that fishery management may have the potential to mitigate the indirect effects of large oil spills when wasp-waist forage fish increase dramatically from an already abundant state following a large and abrupt decrease in the predators that consume them caused by mortality from direct oiling. Perhaps fishery management could mitigate such increases in forage fish by encouraging increased harvest levels, to compensate for the predation control lost through the reductions of natural predators that consume these forage fish.

The poor physiological condition of Gulf menhaden may also have had adverse effects on the species that consume them. Post-spill menhaden were so deficient in fish oil content, and by extension also lipids, fatty acids, and amino acids, that they may have constituted "junk food" [42,54,55]. This adverse effect of the oil spill on food quality diminished the energy content, fish oil, and food value of a dominant forage fish that supports numerous seabirds, marine mammals, predatory fishes and fisheries in the northern GoM ecosystem.

Our study of the relationship between trophodynamics and forage fish prey quality contributes to understanding the broader ecosystem consequences of the DWH blowout.

Moreover, reduced predation following the widespread mortality of piscivorous seabirds may help explain the collapse of the Pacific herring (*Clupea pallasi*) population 3 years after the 1989 *Exxon Valdez* oil spill. The high recruitment of Pacific herring following this oil spill may have reduced their body condition, perhaps making them susceptible to the diseases that are the proximate cause of the crash. The plausibility of this possible explanation should be explored.

### 4.4. Modeling Implications

Our study results have two important implications for how food web models could be constructed to portray the indirect effects of oil spills and other strong environmental perturbations on valued ecosystem components. First, these models should tailor ecosystem boundaries to the habitats of the dominant species that are vulnerable to perturbation. Some published Ecopath and Ecosim food web models aimed at evaluation of the ecological role of Gulf menhaden [56,57] have assumed a habitat area nearly three times the size of the area occupied by most of the population, which diminishes the apparent strength of the modeled trophic linkages to Gulf menhaden accordingly. When tailored to appropriate ecosystem boundaries (e.g., [9]), these models can be invaluable for anticipating and evaluating oil spill effects on vulnerable species and ecological interactions. Second, the issue of food quality and value to consumer species complicates the construction of adequate trophic ecosystem models by which we might implement a widespread food web basis for ecosystem-based management of fisheries, conservation of wildlife, and restoration of ecosystem damages. Some food web models have used energy units, but where energy values are typically constant for a given species. Consequently, using species-specific energy in place of biomass would not solve the food quality issues reflected in the Gulf menhaden data because menhaden's value to its consumers declines at strongly elevated population densities. Perhaps rendering energy value per individual Gulf menhaden could be done using an inversely density-dependent function of biomass to handle this relationship. However, further study may reveal that deficiencies within consumers in certain specific lipids or amino acids demand that the analyses become much more complex, by characterizing major prey species and their consumers via the suite of particular lipids, $\omega$-fatty acids, or amino acids they contain. Clearly, incorporating the concerns over "junk food" and the varying value of prey fish challenges food web modeling and complicates assessment of the consequences of alternative fisheries management, wildlife conservation, and ecosystem restoration schemes.

**Supplementary Materials:** The following are available online at https://www.mdpi.com/2077-1312/9/2/190/s1: Gulf menhaden stock assessment data; and Figure S1: Condition factor of Gulf menhaden sampled opportunistically during spring (Sp), summer (Su), and fall (Fa), east and west of the Mississippi River, from 2011 through 2014.

**Author Contributions:** Conceptualization, J.W.S., C.H.P., C.M.V., M.L.V., V.G., J.C.H., H.J.G.; methodology, J.W.S., C.M.V., M.L.V.; investigation, J.W.S., C.M.V., M.L.V.; data curation, J.W.S., C.M.V., M.L.V.; writing—original draft preparation, J.W.S., C.H.P.; writing—review and editing, J.C.H., H.J.G., V.G., C.M.V., M.L.V.; visualization, J.W.S.; supervision, J.W.S., C.M.V., M.L.V.; project administration, J.W.S., C.H.P.; funding acquisition, J.W.S. All authors have read and agreed to the published version of the manuscript.

**Funding:** This study was in part funded jointly by the Murray Law Firm and by Cossich, Sumich, Parsiola & Taylor LLC. Findings in this manuscript reflect those of the authors only.

**Institutional Review Board Statement:** Not applicable-the only animals used in this study were Gulf menhaden that had been caught and killed by the commercial fishery.

**Informed Consent Statement:** Not applicable.

**Data Availability Statement:** The data presented in this study are available on request from the corresponding author. These data are not publicly available because no public data repository for them has been created.

**Acknowledgments:** The authors thank J. Allen and C. Brodersen for assistance with preparation of the figures, A. Falcone for sampling logistics support, and R. Heintz and the anonymous reviewers for their helpful comments on earlier manuscript versions. We dedicate this work to the memory of Charles Peterson, who passed away during the final stages of manuscript preparation, and whose guidance was invaluable throughout this project.

**Conflicts of Interest:** Three employees of the Murray Law Firm assisted with length and weight measurements of Gulf menhaden, under the direct supervision of J.W.S., C.M.V. and M.L.V. One employee each of the Murray Law Firm and of Cossich, Sumich, Parsiola & Taylor LLC arranged for sample storage and processing facilities, and maintained chain-of-custody records for Gulf menhaden samples.

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
