# Peer review of "Evidence for Ecosystem-Level Trophic Cascade Effects Involving Gulf Menhaden (Brevoortia patronus) Triggered by the Deepwater Horizon Blowout"

_jmse, doi:10.3390/jmse9020190_

Round 1
Reviewer 1 Report
This manuscript investigates the role of the Menhaden fish population in the northern Gulf of Mexico nearshore ecosystem based on data gathered on individual fish from commercial fishing vessels operated out of the Daybrook Fisheries Inc port and processing facility located at Empire LA in 2011. These data were evaluated in context with long time series from the NMFS.
The study is well-designed and the data analysis methods are appropriate. By placing their data in a historical context, these authors identify declines in the physiological condition of the Menhaden population (fish ages 1-3) in the aftermath of the DWH oil spill and hypothesize on the cascading effects of a decline in physiological condition on coupled components of the ecosystem.
They develop and assess a conceptual model of food web interactions that fits the data. They conclude that food web analyses alone are not sufficient to capture the ecosystem consequences of major oil spills.
While the utility of the conceptual model cannot be fully tested and, may be just as easily attributed to selective interpretation of the data (confirmation bias), it is a compelling hypothesis, worthy of being described, discussed, and defended to the extent possible based on the limitations of the data.
I therefore recommend acceptance of this manuscript after consideration of my detailed comments given in the attached file.

Author Response
Main issues to be considered and addressed toward improving the manuscript:
- Small sample size of the data set: The data set that forms the basis of the manuscript consists of ~5,400 individual fish length and weight measurements each year for 4 years. With an abundance of 10’s of billions of fish (age 1+2) in the Gulf region, the sample sizes obtained for this study are certainly not representative of the population as a whole. The authors should acknowledge the limitations of the data set as it is used to support the findings and the further development of the conceptual model created by these authors.
Response:
For random samples, which we assume these samples approximate, the standard error decreases as the inverse of the square root of the samples size (Thompson 2012). The proportion of the total population that is sampled is essentially irrelevant until a very large fraction of the entire population becomes part of the sample. Importantly, we assume each sample is an approximately random sample of the of the DFI catch in each of the area divisions, which form the statistical populations. Indeed, these statistical populations do form a part of the total biological population prior to harvest, but the fishery catch is the statistical population that we are making inference about.
For our purposes, the percent of the total sample of the statistical population is not important, and it is the number of fish in the sample that is substantially controlling precision. For a point of reference, one sample of size 30, if it was an approximately random sample, would produce a standard error of the sample mean of one over the square root of 30, or 0.18, times the original statistical population standard deviation. However, 60 samples of size 30, if these could be considered approximately random samples, would reduce the standard error of the mean to just 0.024 times the original population standard deviation. This sample size should be adequate to characterize this population. Doubling that sample size at this point would have little effect on the standard error of the mean, and that much larger sample size would produce a standard error of the mean of 0.018 times the original population standard deviation, not substantially different from 0.024.
Finally, our sample size of ~5,400 fish for the DFI port is about half the number of fish sampled annually by NMFS for all ports since 1964 to manage the fishery.
(Thompson, S.K. 2012. Sampling, 3rd ed. John Wiley and Sons.)
- Assessment of food web linkages and effects caused by the DWH oil spill (Section 4.3): The authors describe and discuss the wasp-waist food web concept and the importance of such fish in regulating the populations of their prey species. The authors posit that due to the high abundance of Menhaden in 2010, they may have suppressed recruitment of other species through increased predation on planktonic life stages. While this makes sense intuitively, it may not be true. The authors do not really have strong supporting evidence for this hypothesis. Which then naturally, weakens the entire premise that Menhaden is a wasp-waist species in the northern region of the Gulf. At a minimum, I would like to see the alternative be acknowledged. See for example:
Broch, OJ, Nepstad, R, Ellingsen, I, Bast, R, Skeie, GM, Carroll, J (2020) Simulating crude oil exposure, uptake and effects in North Atlantic Calanus finmarchicus populations, Marine Environmental Research, https://doi.org/https://doi.org/10.1016/j.marenvres.2020.105184.
Response: We acknowledge the possibility of toxic effects of oil on plankton, citing the reference suggested above, in the following sentence added at line 364 of the initially-submitted manuscript:
"A recent modeling study suggests that oil toxicity may reduce zooplankton abundance (Broch et al. 2020), but any such reductions would likely be limited to 2010 when oil contamination of surface waters was widespread in the region."
- Effects of exposure to crude oil: I appreciate the clear assessment and presentation of the potential reasons for negligible crude oil impacts on this species after the oil spill (starting at lines 382) and the further assessment of the black particulates hypothesis. The weight of evidence related to the potential impacts of oil spills is too often based on laboratory experimental study results which can never simulate what happens in a real world setting where organisms and oil are not continuously overlapping in space and time and the composition of the populations are evolving. The dynamics of these in-situ processes and their outcomes need far more attention and acknowledgement of their importance in impact assessment.
The proposed alternate explanation for the herring population crash is also well thought and feasible. Further exploration of this through research in the future, if feasible, will be of great value.
Response: We abbreviated this part of the discussion in response to concerns presented by reviewer #2, but we agree this is worth exploring.
- Model and analysis of evidence (4.4)
Suggest including the following paper as an example of multi-species modeling (line 580-582) as it is aimed specifically at analyzing the impacts on a commercial fish species from major oil spills:
Carroll, J., Vikebø, F., Howell, D., Broch, O.-J., Nepstad, R., Augustine, S., Skeie, G.-M., Bast, R., Juselius, J. (2018). Resiliency of a healthy fish stock to recruitment losses from oil spills. Marine Pollution Bulletin. 126:63-73 https://doi.org/10.1016/j.marpolbul.2017.10.069
Response: We eliminated the first paragraph of the Modelling Implications subsection of the Discussion, where this paper would have most naturally fit, but we included reference to it at line 386 in the third paragraph of the Discussion where we address toxicity effects of the oil spill.
Reviewer 2 Report
This paper uses various data sources to build a case for the Deepwater Horizon oil spill affecting Gulf menhaden indirectly through trophic interactions. There is an interesting hypothesis about the role that Gulf menhaden play in this ecosystem that is evaluated using estimates of menhaden age-dependent mass, size-adjusted body mass, and oil yield. Together with other sources of information from other studies, this paper provides evidence that the oil spill increased menhaden abundance, decreased body mass, and lipid content. These changes in menhaden population size and condition also likely affect many species in the Gulf ecosystem, given the vital role that menhaden play for many predators.
While this paper successfully gathered many sources of information to evaluate this hypothesis, a couple of critical issues reduce the paper’s effectiveness. First, the paper is quite lengthy, and while the hypothesis is complicated to answer, some sections are superfluous to the core question. In particular, the discussion has some speculative digressions into related research in other ecosystems that are not the paper’s primary focus. Additionally, some concepts are laboriously described and could be communicated more quickly. See my line-by-line comments for more specific suggestions, but this paper could be a lot shorter than it is.
As for the analysis, while I do not think that research papers need to be devoted to p-values, I think that variance in response variables need to be accounted for to make a reasonable inference. While there was some effort to characterize differences in response variables after the spill to overall averages, uncertainty in annual estimates was not accounted for in many aspects of the study. Even if uncertainty is challenging (or impossible) to estimate, more effort is needed to describe its effect on the inference made in this paper. This paper desibes a compelling story, but more work is needed for the article to be effective and convincing.
Line-by-line comments:
39: ‘Lower levels’ of organization is relative and unclear about which aspects of the trophic system are referenced here.
45: The sentence ending on this line needs a reference. Fox et al. (2018) or Haney et al. (2014) would be reasonable.
75: There is some cool work looking at the effect of menhaden on seabird populations from Lamb et al. (2017) that is worth mentioning here.
77-84: While this example helps describe ‘wasp-waist’ forage species, this section could essentially just be the last sentence, and the reader will know what you are describing.
85: The paragraph starting here is useful information, but it doesn’t seem necessary to communicate the paper. Even if this information were needed, it would probably be better served in the discussion after the audience has been shown this study’s findings. Additionally, this section supports one of the more speculative portions of the discussion, which is unneeded as the paper does not directly evaluate the Valdez oil spill.
156: Are the results from DFI indicative of the entire Gulf menhaden population? Their catch represents 26-43% of the total sample. Is that enough to be representative?
207: Equation 1 is an excellent example of a point estimate that needs uncertainty incorporated into it. It’s challenging to interpret values like these without describing the variation inherent to population size or body mass.
218: Table 1 should be two different tables. It’s odd seeing columns change their meaning as we move from A to B.
235: Is this a reasonable assumption to make? What evidence is there that this is the case? If it isn’t possible to provide evidence for the assumption, how important is it to estimating productivity?
251: How much does this estimate of trophic transfer efficiency matter to the results found here? The estimate of the parameter comes from a while back, and I wonder if we have a good sense of how it affects the current results.
253: Are these estimates of trophic level informing this analysis at all? This sentence struck me as out of place.
261: This is one of the few times that differences in the data were contextualized with statistics. More documentation of the long-term mean, it’s standard deviation and the relative difference between 2010-2012 data and the global mean would be useful. However, this analysis ignores uncertainty in the annual estimates themselves. While incorporating this uncertainty would not affect the size of the difference, the precision with which these values were estimated is essential to interpreting the difference. As this stands, I don’t know if a difference of this magnitude can be effectively estimated in this study, which makes the result difficult to believe.
269: Again, this analysis was conducted just by comparing values without uncertainty estimates. This time, the metric of standard deviations away from the mean wasn’t even used to describe how large an aberration the points were. The assessment of the pattern is fine, but I have no idea how noisy all these data are.
274: This seems to be a written description of the figure, and it doesn’t need to be this long. Let the figure tell the story.
286: Figure 4 includes an estimate of Mississippi discharge that isn’t even discussed in the results. It’s referenced later in the discussion, but the lack of reference to this in the text makes this figure too confusing. Some discussion of that part of the figure is needed here.
316-318: This is the core takeaway of this paragraph that the reader needs to understand. The restate just restates the methods and describes Tables 1 and 2 again.
325-334: Shouldn’t this just be in the Methods?
346-348: This final sentence is what the reader needs to understand. The rest of the paragraph takes too long to get to this point and is too long.
367-369: I found it strange that the only statistical analysis of anything presented in this paper is described in the Discussion and is from another article. I appreciate the correlation coefficient, p-value, and degrees of freedom, but expecting the readers of this paper to understand that analysis well enough to interpret this result is probably asking for too much.
414-417: Climate change affecting Gulf menhaden habitat is another reason why the assumption on line 229. If the effect of climate is large, it seems like the estimates of productivity could be biased.
450-454: Given that the 15.4% estimate is a minimal one, how much uncertainty is introduced from the unmeasured sources of production in the early age classes? You suggest that it might be as much as 5% or a full 25% of the unbiased total. That seems is a fair bit of variation explained by those age classes and could result in biasing the annual estimates you are relying on to make your argument about the oil spill’s effects.
479-480: Another good spot for the Lamb et al. (2017) paper.
502: This paragraph is speculative. I think it’s fair to suggest a trophic cascade is occurring, but you don’t have any direct evidence for it in this study. I would cut back on this section a bit, though the suggestion that reductions in zooplankton contribute to the hypoxic zone was interesting.
547: Again, this paragraph is speculative. You might be right, and I think it’s valid to suggest this mechanism for the observation seen during the Valdez spill but spending an entire paragraph on this speculation gives it more validity than it deserves.
565: I would cut this entire section. This discussion is outside the scope of the paper. I think it is reasonable to suggest that food-web models can inform various aspects of species or ecosystem modeling. And it certainly appears that some models need to consider the findings of studies like this. But this discussion point requires a lot of explanation to get the readers to understand what you are talking about, and the recommendations that are made still need more explanation than they already do. It seems like summarizing the key recommendations in a conclusions section would serve the paper much better.
513: Figure 5 does a nice job of walking the reader through the top-down trophic mechanisms being proposed, but it doesn’t account for other sources of variation in the food supply. Potential mechanisms like freshwater input are mentioned in the paper but they don’t make it into the diagram here which is very focused on top-down effects.
Literature Cited
Fox, C.H., O'hara, P.D., Bertazzon, S., Morgan, K., Underwood, F.E. and Paquet, P.C., 2016. A preliminary spatial assessment of risk: Marine birds and chronic oil pollution on Canada's Pacific coast. Science of the Total Environment, 573, pp.799-809.
Haney, J.C., Geiger, H.J. and Short, J.W., 2014. Bird mortality from the Deepwater Horizon oil spill. I. Exposure probability in the offshore Gulf of Mexico. Marine Ecology Progress Series, 513, pp.225-237.
Lamb, J.S., Satgé, Y.G. and Jodice, P.G., 2017. Diet composition and provisioning rates of nestlings determine reproductive success in a subtropical seabird. Marine Ecology Progress Series, 581, pp.149-164.
Author Response
General comment regarding paper length:
Response: Unfortunately our attempts to address the concerns of this reviewer have offset the considerable reductions in length we have made elsewhere, so the paper remains about as long as the previous version. But we have removed hopefully all of the text that this reviewer found speculative, digressive, or inefficiently communicated, closely following the reviewer's advice in the line-by-line comments.
General comment regarding statistical analysis:
Response: We greatly appreciate this reviewer's pointing out this glaring oversight. We have added a subsection to the methods that addresses data quality and parameter uncertainty, and we have indicated the 95% confidence intervals for all the data points presented in Fig. 2 and for the condition index data in Fig. 3.
Line-by-line comments:
39: ‘Lower levels’ of organization is relative and unclear about which aspects of the trophic system are referenced here.
Response: We replaced "lower levels" with "the organismal or population levels" here.
45: The sentence ending on this line needs a reference. Fox et al. (2018) or Haney et al. (2014) would be reasonable.
Response: We added (National Research Council 2003) here, as it includes reference to marine mammals in addition to seabirds, and effects on intertidal organisms.
75: There is some cool work looking at the effect of menhaden on seabird populations from Lamb et al. (2017) that is worth mentioning here.
Response: We added this citation here.
77-84: While this example helps describe ‘wasp-waist’ forage species, this section could essentially just be the last sentence, and the reader will know what you are describing.
Response: We removed the following text here:
"The term “wasp-waist” refers to a food web in which most of the lower trophic level production of the diverse plankton flows through very few, often one or two, species of forage fishes that in turn support a diversity of predators. High species richness at trophic levels immediately above and below that of one or two forage fish species as depicted on food web diagrams is analogous in graphical shape to the waist connecting the much larger abdomen and thorax of a wasp."
85: The paragraph starting here is useful information, but it doesn’t seem necessary to communicate the paper. Even if this information were needed, it would probably be better served in the discussion after the audience has been shown this study’s findings. Additionally, this section supports one of the more speculative portions of the discussion, which is unneeded as the paper does not directly evaluate the Valdez oil spill.
Response: We removed this paragraph.
156: Are the results from DFI indicative of the entire Gulf menhaden population? Their catch represents 26-43% of the total sample. Is that enough to be representative?
Response:
We do not believe there is need to assume that the catch is exactly representative of the population. If there is a sudden and dramatic change in the characteristics of fish in the catch, temporally or spatially, that would necessarily indicate a corresponding change in the population that supports the catch, whether the magnitude of the two changes were exactly the same or not. We added wording near line 159 to make this point.
207: Equation 1 is an excellent example of a point estimate that needs uncertainty incorporated into it. It’s challenging to interpret values like these without describing the variation inherent to population size or body mass.
Response: We addressed this issue in a new subsection we added to the Methods entitled "Data Quality and Parameter Uncertainty".
218: Table 1 should be two different tables. It’s odd seeing columns change their meaning as we move from A to B.
Response: We split this table into two separate tables. The new Table 1 contains only the weight-at-age data, and the new Table 2 contains the abundance-at-age and productivity data. In addition, we added a column to indicate biomass in Table 2, and another column to indicate the ratio of production to biomass, to make the basis for the ratio of production to biomass noted in the results and discussion more accessible to the reader. These biomass estimates are ~10% higher than those presented in Short et al. 2017, because biomass computed using the new eq. 1 is for the beginning of the calendar year when the biomass is near its annual peak. Biomass is near the annual minimum in July as a result of removals by the fishery that are not fully compensated by the growth of the age-0 cohort.
235: Is this a reasonable assumption to make? What evidence is there that this is the case? If it isn’t possible to provide evidence for the assumption, how important is it to estimating productivity?
Response: We added the following text to address these concerns:
"This assumption is justified by the facts that most of the Gulf menhaden population resides within the 20 m isobath (Turner 1969), most of the estuarine nursery habitat is along the coasts of Louisiana and Mississippi (Schueller et al. 2013), Gulf menhaden tend to migrate towards the Mississippi River delta as they age (Ahrenholz 1981), and nearly all of the commercial fishing grounds are located within the 20 m isobath from Galveston Bay, Texas to Mobile Bay, Alabama (see Fig. 5.11 in Schueller et al. 2013)."
251: How much does this estimate of trophic transfer efficiency matter to the results found here? The estimate of the parameter comes from a while back, and I wonder if we have a good sense of how it affects the current results.
Response: We addressed this issue in a new subsection we added to the Methods entitled "Data Quality and Parameter Uncertainty".
253: Are these estimates of trophic level informing this analysis at all? This sentence struck me as out of place.
Inserted "i =" before 2.63 and 3.46 on line 254, and added "..., and are used in eq. 3 to estimate total productivity at trophic level i" at the end of this sentence on line 255. We also provide examples of the critical role of these estimates of trophic level in estimating the proportion of net primary productivity that flows through age-0 and age-1+ Gulf menhaden in the data quality and parameter estimation subsection of the Methods.
261: This is one of the few times that differences in the data were contextualized with statistics. More documentation of the long-term mean, it’s standard deviation and the relative difference between 2010-2012 data and the global mean would be useful. However, this analysis ignores uncertainty in the annual estimates themselves. While incorporating this uncertainty would not affect the size of the difference, the precision with which these values were estimated is essential to interpreting the difference. As this stands, I don’t know if a difference of this magnitude can be effectively estimated in this study, which makes the result difficult to believe.
Response: We addressed this issue in the data quality and parameter uncertainty subsection, and by noting the small size of the 95% confidence intervals for the data presented in Figs. 2 and 3.
269: Again, this analysis was conducted just by comparing values without uncertainty estimates. This time, the metric of standard deviations away from the mean wasn’t even used to describe how large an aberration the points were. The assessment of the pattern is fine, but I have no idea how noisy all these data are.
Response: We addressed this issue in the data quality and parameter uncertainty subsection, and by noting the small size of the 95% confidence intervals for the data presented in Figs. 2 and 3.
274: This seems to be a written description of the figure, and it doesn’t need to be this long. Let the figure tell the story.
Response: We removed the last four sentences of this paragraph.
286: Figure 4 includes an estimate of Mississippi discharge that isn’t even discussed in the results. It’s referenced later in the discussion, but the lack of reference to this in the text makes this figure too confusing. Some discussion of that part of the figure is needed here.
Response: We added "During this period (2011 through 2014), Mississippi River discharge during May was unusually high in 2011, unusually low in 2012, and near average in 2013 and 2014 (Fig. 4)."
316-318: This is the core takeaway of this paragraph that the reader needs to understand. The restate just restates the methods and describes Tables 1 and 2 again.
Response: We condensed this paragraph. However, the data that remain are crucial for readers who wish to follow our arguments quantitatively.
325-334: Shouldn’t this just be in the Methods?
Response: We re-wrote this paragraph. While some of this material could be in the methods (where some of it already is), the presentation here continues the thread set up beginning on line 305, and we believe moving much of this to the Methods would be more confusing.
346-348: This final sentence is what the reader needs to understand. The rest of the paragraph takes too long to get to this point and is too long.
Response: As noted in our response to the previous comment, we re-wrote this paragraph.
367-369: I found it strange that the only statistical analysis of anything presented in this paper is described in the Discussion and is from another article. I appreciate the correlation coefficient, p-value, and degrees of freedom, but expecting the readers of this paper to understand that analysis well enough to interpret this result is probably asking for too much.
Response: The important overarching conclusion we want the reader to take away is that based on several lines of inquiry: there was a sudden and dramatic change in the Gulf menhaden populations affected by the Deepwater Horizon oil spill. Where we could, we did offer support for our conclusions based on observed variation. For example, we noted a sudden and dramatic in the weight at age for age-1 fish at Empire port (near line 261). We supported our conclusion that this was indeed a change with the evidence that (1) this was the lowest weight on record, and (2) that it was more than 3 standard deviations below the long-term mean. One could make some assumptions about the distribution of the values and then develop a statistical hypothesis test. But reporting how many standard deviations away from the long-term mean the value was conveys exactly the same information without the need unnecessary assumptions. Someone who stylistically prefers p-values can refer to a table and note that 3 standard deviations will represent a p-value of approximately 0.0014—after making some assumptions we simply chose not to make. The strongest evidence for our overarching conclusion was not simply a hypothesis test. Rather our evidence is the redundant signals showing reduced oil content, reduced condition factor, and extreme recruitment all coherently observed in time and space in different high-quality fishery metrics.
414-417: Climate change affecting Gulf menhaden habitat is another reason why the assumption on line 229. If the effect of climate is large, it seems like the estimates of productivity could be biased.
Response: Gulf menhaden production shows no evidence of trend from 2000 to 2009 in our Table 2. While higher water temperatures would be expected to increase the metabolic rate of poikilotherms generally, reference to the Q10 relationship suggests that a temperature increase of perhaps 0.2 °C over the decade of the 2000s might increase metabolic rates by around 2%. This is considerably smaller than the uncertainty of our productivity estimates, and also much smaller that the production increases in 2011 and 2012 that we attribute to increased Gulf menhaden recruitment.
Also, we addressed likely effects of climate change in the paragraph that began on line 401 of the initially-submitted manuscript.
450-454: Given that the 15.4% estimate is a minimal one, how much uncertainty is introduced from the unmeasured sources of production in the early age classes? You suggest that it might be as much as 5% or a full 25% of the unbiased total. That seems is a fair bit of variation explained by those age classes and could result in biasing the annual estimates you are relying on to make your argument about the oil spill’s effects.
Response: We have no basis for speculating as to the uncertainty associated with the productivity of age-0 Gulf menhaden, and of age-1+ Gulf menhaden that escape the fishery. We agree that it must be a substantial source of productivity. Our fundamental point here is that this independent approach to estimating the proportion of net primary productivity appropriated by Gulf menhaden is broadly consistent with the approach based on eq. 1 and our assumption that 75% of the entire population resides within the blue shaded area on Fig. 1.
479-480: Another good spot for the Lamb et al. (2017) paper.
Response: We added the Lamb paper to the references cited here.
502: This paragraph is speculative. I think it’s fair to suggest a trophic cascade is occurring, but you don’t have any direct evidence for it in this study. I would cut back on this section a bit, though the suggestion that reductions in zooplankton contribute to the hypoxic zone was interesting.
Response: We removed this sentence: " This reflects the logical extension of a trophic cascade initiated by the losses of piscivorous seabirds and marine mammals from direct oiling (Haney et al. 2014, Schwacke et al. 2014, Venn-Watson et al. 2015), and reduced access of stenohaline predators to juvenile Gulf menhaden resulting from the freshwater diversions after the DWH blowout (Fig. 5).", and added a clause to the preceding sentence "..., suggesting the possibility of trophically-cascading effects."
547: Again, this paragraph is speculative. You might be right, and I think it’s valid to suggest this mechanism for the observation seen during the Valdez spill but spending an entire paragraph on this speculation gives it more validity than it deserves.
Response: We condensed this paragraph to: " Our study of the relationship between trophodynamics and forage fish prey quality contributes to understanding the broader ecosystem consequences of the DWH blowout. Moreover, reduced predation following widespread mortality of piscivorous seabirds may help explain the collapse of the Pacific herring (Clupea pallasi) population 3 years after the 1989 Exxon Valdez oil spill. High recruitment of Pacific herring following this oil spill may have reduced their body condition, perhaps making them susceptible to the diseases that are the proximate cause of the crash. The plausibility of this possible explanation should be explored."
565: I would cut this entire section. This discussion is outside the scope of the paper. I think it is reasonable to suggest that food-web models can inform various aspects of species or ecosystem modeling. And it certainly appears that some models need to consider the findings of studies like this. But this discussion point requires a lot of explanation to get the readers to understand what you are talking about, and the recommendations that are made still need more explanation than they already do. It seems like summarizing the key recommendations in a conclusions section would serve the paper much better.
Response: This paper will appear in a special issue of JMSE devoted to oil spill modeling, so we feel compelled to address implications of our study for such modelling. However, we agree that the first paragraph of this section could be omitted, and have done so.
513: Figure 5 does a nice job of walking the reader through the top-down trophic mechanisms being proposed, but it doesn’t account for other sources of variation in the food supply. Potential mechanisms like freshwater input are mentioned in the paper but they don’t make it into the diagram here which is very focused on top-down effects.
Response: The function of this figure is to summarize the top down effects we propose, not to summarize all of the factors that might modulate Gulf menhaden condition or abundance.
References added (2):
Dickinson, W.T.; Accuracy of discharge determinations; Hydrology paper #20, Colorado State University, Fort Collins, USA. 1967.
Österblom, H.; Casini, M,; Olsson, O.; Bignert, A. Fish, seabirds and trophic cascades in the Baltic Sea. Mar. Ecol. Prog. Ser. 2006, 323, 233-238.
References removed in revised submission (16):
Ainsworth, C.H. Strategic marine ecosystem restoration in Northern British Columbia. Ph.D. Thesis, University of British Columbia, Vancouver, British Columbia, May 2006.
Bishop, M.A.; Watson, J.T.; Kuletz, K.; Morgan, T. Pacific herring (Clupea pallasii) consumption by marine birds during winter in Prince William Sound, Alaska. Fish. Oceanogr. 2015, 24, 1-13.
Botsford, L.W.; Castilla, J.C.; Peterson, C.H. The management of fisheries and marine ecosystems. Science 1997, 227, 509-515.
Christensen, V.; Walters, C.J. Ecopath with Ecosim: methods, capabilities and limitations. Ecol. Model. 2004, 172, 109-139.
Frost, K.; Lowry, L.; Sinclair, E.; Ver Hoef, J.; McAllister, D. Impacts on distribution, abundance and productivity of harbor seals; In Marine Mammals and the Exxon Valdez; Loughlin, T., Ed.; Academic Press: San Diego, USA, 1994; pp. 97-118.
Herschberger, P.K.; Kocan, R.M.; Elder, N.E.; Meyers, T.R.; Winton, J.R. Epizootiology of viral hemorrhagic septicemia virus in Pacific herring from the spawn-on-kelp fishery in Prince William Sound, Alaska, USA. Dis. Aquat. Org., 1999, 37, 23-31.
Marty, G.D.; Freiberg, E.F.; Meyers, T.R.; Wilcock, J.; Farver, T.B.; Hinton, D.E. Viral hemorrhagic septicemia virus, Ichyophonus hofen, and other causes of morbidity in Pacific herring, Clupea pallasi, spawning in Prince William Sound. Dis. Aquat. Org., 1988, 32, 15-40.
Okey, T.; Pauly, D. A mass-balanced model of trophic flows in Prince William Sound: decompartmentalizing ecosystem knowledge. In Ecosystem Approaches for Fisheries Management 621, AK-SG-99-01; Alaska Sea Grant College Program: Fairbanks, Alaska, USA, 1999; pp. 621-635
Pearson, W.H.; Deriso, R.B.; Elston, R.A; Hook, S.E.; Parker, K.R.; Anderson, J.W. Hypotheses concerning the decline and poor recovery of Pacific herring in Prince William Sound, Alaska. Rev. Fish Biol. Fisheries, 2012, 22, 95-135.
Peterson, C.H.; Grabowski, J.H.; Powers, S.P. Estimated enhancement of fish production resulting from restoring oyster reef habitat: quantitative valuation. Mar. Ecol. Prog. Ser., 2003, 264, 251-266.
Piatt J.F.; Lensink, C.J. Exxon Valdez bird toll. Nature, 1989, 342, 865-866.
Pitkitch, E.K.; Santora, C.; Babcock, E.A.; Bakun, A.; Bonfil, R.; Conover, D.O.; Dayton, P.; Doukakis, P.; Fluharty, D.; Heneman, B.; Houde, E.D.; Link, J.; Livingston, P.A.; Mangel, M.; McAllister, M.K.; Pope, J.; Sainsbury, K.J. Ecosystem-based fishery management. Proc. Natl. Acad. Sci. U.S.A., 2004, 305, 346-347.
Ruckelshaus, M.; Klinger, T.; Knowlton, N.; DeMaster, D.P. Marine ecosystem-based management in practice: scientific and governance challenges. BioScience, 2008, 58, 53-63.
Schwacke, L.H.; Smith, C.R.; Townsend, F.I.; Wells, R.S.; Hart, L.B.; Balmer, B.C.; Collier, T.K.; De Guise, S.; Fry, M.M.; Guillette, L.J.Jr.; Lamb, S. V. Health of common bottlenose dolphins (Tursiops truncatus) in Barataria Bay, Louisiana, following the “Deepwater Horizon” oil spill. Environ. Sci. Technol., 2014, 48, 93-103.
Travis, J.; Coleman, F.C.; Auster, P.J.; Cury, P.M.; Estes, J.A.; Orensanz, J.; Peterson, C.H.; Power, M.E.; Steneck, R.S.; Wootton, J.T. Integrating the invisible fabric of nature into fisheries management. Proc. Natl. Acad. Sci. U.S.A., 2014, 111, 581-584.
Venn-Watson, S.; Colegrove, K.M.; Litz, J.; Kinsel, M.; Terio, K.; Saliki, J.; Fire, S.; Carmichael, R.; Chevis, C.; Hatchett, W.; Pitchford, J. Adrenal gland and lung lesions in Gulf of Mexico Common Bottle-nose Dolphins (Tursiops truncatus) found dead following the “Deepwater Horizon” oil spill. PLOS ONE 2015, 10, e0126538.
Round 2
Reviewer 2 Report
Thank you for your well-considered responses to my suggestions. This version of the paper is much improved and I appreciate the thorough description of your analytical uncertainties.